# Fragility of surface states in topological superfluid $^3$He

P. J. Heikkinen [1✉], A. Casey [1], L. V. Levitin [1], X. Rojas [1], A. Vorontsov [2], P. Sharma [3], N. Zhelev [4], J. M. Parpia [4] & J. Saunders[1✉]

Superfluid $^3$He, with unconventional spin-triplet p-wave pairing, provides a model system for topological superconductors, which have attracted significant interest through potential applications in topologically protected quantum computing. In topological insulators and quantum Hall systems, the surface/edge states, arising from bulk-surface correspondence and the momentum space topology of the band structure, are robust. Here we demonstrate that in topological superfluids and superconductors the surface Andreev bound states, which depend on the momentum space topology of the emergent order parameter, are fragile with respect to the details of surface scattering. We confine superfluid $^3$He within a cavity of height $D$ comparable to the Cooper pair diameter $\xi_0$. We precisely determine the superfluid transition temperature $T_c$ and the suppression of the superfluid energy gap, for different scattering conditions tuned in situ, and compare to the predictions of quasiclassical theory. We discover that surface magnetic scattering leads to unexpectedly large suppression of $T_c$, corresponding to an increased density of low energy bound states.

[1] Department of Physics, Royal Holloway University of London, Surrey, UK. [2] Department of Physics, Montana State University, Bozeman, MT, USA.
[3] Department of Physics, Indian Institute of Science, Bangalore, India. [4] Department of Physics, Cornell University, Ithaca, NY, USA. ✉email: petri.heikkinen@rhul.ac.uk;
j.saunders@rhul.ac.uk

The spin-triplet superfluid phases[1–3] of liquid $^3$He provide a benchmark for topological superconductivity[4]. So far such superconductors[5,6] are the missing 'elements' in the periodic table of quantum matter[7]; while candidates exist, such as $Sr_2RuO_4$[8], $UPt_3$[9], doped $Bi_2Se_3$[10], $UTe_2$[11], no bulk material has yet been unambiguously identified as a topological crystalline superconductor[12]. Nevertheless, devices fabricated from spin-triplet crystalline topological superconductors should eventually contribute to potential applications in topological quantum computation[13–15]. Current strategies explore the manipulation of Majorana zero modes (MZMs) in devices which rely on 1-D topological superconductivity[16–19] induced by the proximity effect in topological insulators or semiconductors with strong spin–orbit coupling, or via planar Josephson Junctions[20,21]. However, MZMs may be corrupted by non-topological low energy Andreev bound states (ABS)[22], which can be present as a result of details of interfaces and materials properties in such systems[23]. Here we report the fragility of surface ABS in superfluid $^3$He at an ideal non-transparent interface, exploiting the ability to tune in situ the scattering of quasiparticles by the surface through adjustment of the isotopic composition of the helium surface boundary layer.

Recently we have shown that it is possible to cool $^3$He confined within precisely engineered nanoscale cavities into the superfluid phases[24], and detect the nuclear magnetic resonance (NMR) response of the small $^3$He sample using an ultra-sensitive spectrometer[25]. Surface scattering dominates the properties under strong confinement. NMR determines the superfluid transition temperature, the pairing state, and the superfluid energy gap[24,26,27], the suppression of which under confinement self-consistently determines, through quasiclassical theory[28,29], the density of states of the ABS mid-gap surface excitations.

Superfluid $^3$He consists of spin-triplet Cooper pairs, with one unit ($l = 1$) of orbital angular momentum[1]. The order parameter is a complex $3 \times 3$ matrix, encoding the spin state of the pairs over the spherical Fermi surface. In bulk liquid, two phases are found with distinct symmetries[1,2] and momentum-space topologies[4]. The A phase is chiral, breaking time-reversal symmetry. Over the Fermi surface, pairs form with the same direction of their orbital angular momentum and in an equal-spin state comprising just $|\uparrow\uparrow\rangle$ and $|\downarrow\downarrow\rangle$ pairs. The B phase is time-reversal invariant, comprising all three components of the spin triplet, with broken relative spin–orbit symmetry. The relative stability of these phases is profoundly altered by confinement[24]. Superfluid $^3$He is an intrinsically impurity-free system, although impurities can be artificially introduced using silica aerogels of different porosities and structure factors[30]. Our study of superfluid $^3$He confined in a simple slab geometry determines the influence of surface scattering alone on gap suppression, in the absence of impurity scattering.

In the quasiclassical theory of superfluid $^3$He[29] the effect of the surface can be characterized in terms of a single parameter, $S$, the degree of specularity of the surface scattering of quasiparticles[31]. There is prior compelling evidence from hydrodynamic and transverse acoustic impedance studies of normal state $^3$He that the surface scattering may be tuned in situ from diffuse to specular[32–34] by coating surfaces with $^4$He. Approaching close to full specularity was found to require superfluidity of the $^4$He surface film[33,35]. In early work it was shown that both the NMR frequency shift of superfluid $^3$He confined in a stack of mylar sheets[36] and the superfluid fraction within the pores of packed powders[37] depend on the surface $^4$He coverage. They both increased with increased coverage, indicating the expected reduced pair breaking with higher specularity. Transverse acoustic impedance measurements, and quasiclassical analysis, revealed in superfluid $^3$He-B the energy density of states of mid-gap surface-bound excitations depends on specularity[38–41]. Their spectrum shows a Majorana-like cone as the conditions for fully specular scattering are approached.

Quasiclassical theory predicts for the A phase that $T_c$ and gap suppression, and surface-bound states are all eliminated for fully specular scattering. Diffuse scattering leads to a finite density of surface-bound states at low energy. In this case, the suppression of $T_c$ scales as $\delta T_c / T_{c0} \propto -(\xi_0/D)^2$ to leading order. In the work reported here, we use NMR to study the A phase in a single cavity of precisely defined dimensions, with several different surface boundary layers to tune the surface quasiparticle scattering.

## Results

**Experimental details.** In our experiment, superfluid $^3$He was confined within a 192 nm high cavity defined in a silicon wafer, Fig. 1a. The effective confinement can be varied at fixed cavity height $D$ by changing the sample pressure and hence the superfluid coherence length $\xi_0 = \hbar v_F / 2\pi k_B T_{c0}$, where $v_F$ is the Fermi velocity and $T_{c0}$ the bulk superfluid transition temperature. Measurements were made at a series of pressures from 0.0 to 5.5 bar, over which $\xi_0$ decreases from 77 to 40 nm. We determine the shift in the NMR resonance frequency $f$ relative to the Larmor frequency $f_L$, $\Delta f = f - f_L$, which occurs in the superfluid state. The onset of this shift identifies $T_c$ in the cavity. This is determined precisely relative to $T_{c0}$ by also observing frequency shifts in small volumes of bulk liquid incorporated in the cell design, Fig. 1a and Supplementary Fig. 1. The frequency shifts from the cavity and the bulk markers are of opposite sign, Fig. 1b, c. The superfluid transition within the cavity is sharp, due to the uniformity of cavity height, relative to that achieved in stacked multiple films with a broad distribution of thickness[36]. The transition temperature depends on whether the surface boundary layer is solid $^4$He or superfluid $^4$He, which determines surface specularity, Fig. 1d. Suppression of the gap by confinement is inferred from the magnitude of the cavity signal frequency shift.

The relatively strong confinement in the 192 nm cavity stabilizes the A phase at all temperatures and pressures, consistent with the phase diagram determined in previous work[24,36,42]. The orbital angular momentum of the pairs, which defines the orientation of point nodes of the gap in momentum space, lies normal to the cavity surface $\hat{\mathbf{l}} = \pm \hat{\mathbf{z}}$. The order parameter $\Delta(\mathbf{p}) = \Delta_A(z)(\hat{p}_x + i\hat{p}_y)[|\uparrow\uparrow\rangle + |\downarrow\downarrow\rangle]$[42], where $\mathbf{p}$ is the Fermi surface momentum, and $z$ is the position across the cavity. Here $\Delta_A$ is the A-phase gap maximum at the Fermi surface equator. In general, the gap has a spatial dependence across the cavity, $\Delta_A(z)$. The static magnetic field $\mathbf{H}_0 = H_0 \hat{\mathbf{z}}$ orients the spins along $\hat{\mathbf{z}}$, via the anisotropic magnetic susceptibility. With this relative orientation of spin and orbital angular momentum, the dipolar energy is maximized, accounting for the negative frequency shift observed from the cavity (Fig. 1 and Supplementary Note 1).

**Confinement with $^4$He surface plating.** We first report measurements in which the sample walls and heat exchanger surfaces were plated with sufficient $^4$He to displace the magnetic solid $^3$He surface boundary layer, which arises naturally in pure $^3$He samples[43]. The plating procedure, which uses a $^4$He surface coverage of 32 μmol m$^{-2}$, results in a non-magnetic localized solid $^4$He surface boundary layer ('Methods'). In this case, the observed $T_c$ suppression is close to that predicted for purely diffuse scattering. Details of the treatment of surface scattering used in our quasiclassical computations are given in Supplementary Note 3. The results are best fit with specularity $S = 0.1$, referred to here as 'diffuse', Fig. 2a. The increase in $T_c$ suppression with decreasing pressure arises naturally from stronger effective confinement[42], Fig. 2b. Subsequently, a

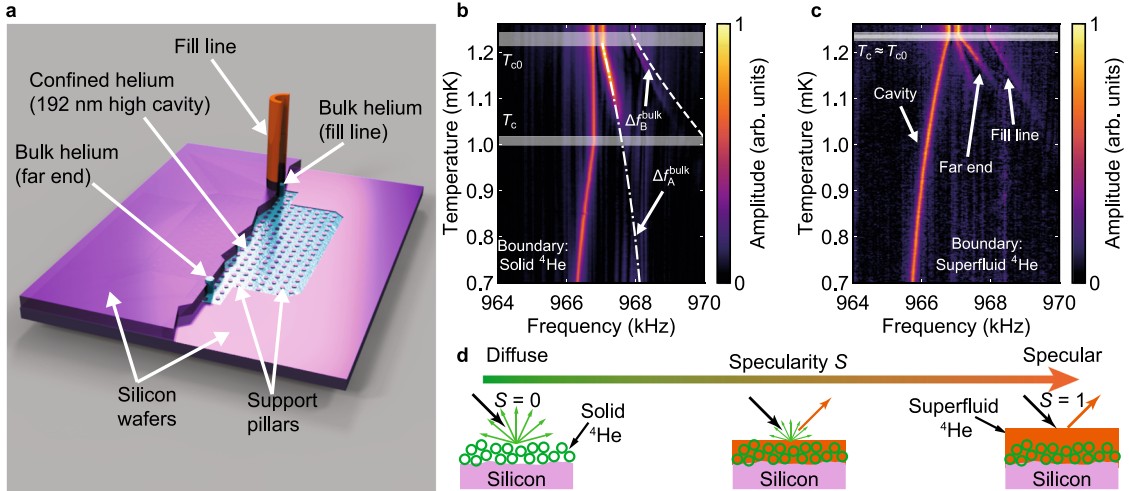

**Fig. 1 Experimental cell confining $^3$He. a** Nanofabricated sample cell, cut away to show cavity in lower silicon wafer, bonded to upper wafer. The support posts shown maintain cavity height $D$ under different liquid pressures. The cavity is filled through a fill line via a sintered heat exchanger and cooled through the column of $^3$He within it. Small volumes of bulk liquid at each end of the cavity provide markers for the bulk superfluid transition $T_{c0}$, and eliminate thermometry errors due to temperature gradients. The NMR coil set around the sample is shown in Supplementary Fig. 1. Suitable small magnetic field gradients are used to resolve the NMR response of different regions of the cell, see 'Methods'. **b, c** NMR signatures of superfluid transition in cavity and bulk markers, for two different surface boundary conditions, at $^3$He pressure of 2.46 bar. $^3$He-A in the cavity shows a negative frequency shift whereas the bulk markers show positive frequency shift; $\Delta f_A^{bulk}$ and $\Delta f_B^{bulk}$ refer to the calculated bulk superfluid frequency shifts of $^3$He-A and $^3$He-B, respectively, see Supplementary Note 2. The $T_c$ suppression observed with a surface boundary layer of solid $^4$He is eliminated by the addition of $^4$He to create a superfluid $^4$He film at the surface. The white horizontal bands show the measured $T_c$ and $T_{c0}$ including the uncertainties in temperature determination, see 'Methods'. **d** Schematic illustration of the tuning of surface scattering conditions, parametrised by specularity coefficient $S$, by surface plating atomically smooth silicon with a $^4$He film. The green circles represent $^4$He atoms, the orange is liquid $^4$He, and the arrows indicate the flux of incoming and outgoing $^3$He quasiparticles.

thicker $^4$He film was formed on the cavity walls to create a surface superfluid film of $^4$He ('Methods'). In this case, we observe an almost complete elimination of $T_c$ suppression, demonstrating close to fully specular scattering, referred to here as 'specular', Fig. 2a, b.

In general, the measured frequency shift is related to the spatial average of the suppressed gap within the cavity via $\left| f^2 - f_L^2 \right| = \zeta \langle \Delta_A^2(z) \rangle$, where $\zeta$ is an intrinsic material parameter which is pressure dependent but temperature independent (Supplementary Note 1). In the Ginzburg–Landau regime, sufficiently close to $T_{c0}$, the A-phase bulk gap maximum $\Delta_A$ is given by $\Delta_A^2 = \frac{\Delta C_A}{C_n} (\pi k_B T_{c0})^2 (1 - T/T_{c0})$, where $\Delta C_A/C_n$ is set to the measured heat capacity jump at $T_{c0}$. This expression thus incorporates strong-coupling corrections to the gap near $T_{c0}$ (Supplementary Note 4). For 'specular' boundaries, the measured cavity frequency shift corresponds to the unsuppressed bulk gap, Fig. 2c, and allows determination of the constant $\zeta$ at each pressure. For the 'diffuse' boundary, using the determined value of $\zeta$, we can precisely infer the gap suppression from the measured frequency shift, independent of uncertainties in material parameters and temperature scale, Supplementary Note 1. We find that the observed gap suppression is also best described by $S = 0.1$, establishing the consistency of the experimentally determined gap suppression and $T_c$ suppression within the framework of quasiclassical theory.

**Confinement without $^4$He surface plating**. We now turn to the results where no $^4$He preplating was deployed. Rapid exchange between the magnetic surface boundary layer of localized $^3$He and the liquid results in a single hybridized NMR line[36]. The superfluid transition temperature is inferred from analysis of the frequency shift of the hybridized line, which is a weighted average of the internal dipolar frequency shift in the solid $^3$He surface boundary layer and that due to superfluidity[36] (Supplementary

Note 5). It shows an unexpectedly large $T_c$ suppression, Fig. 3a, significantly exceeding that observed with a solid $^4$He boundary layer, and inconsistent with diffuse scattering $S \approx 0$. This result can be phenomenologically described in terms of an effective specularity $S_{eff} = -0.4$. This approaches the condition for maximal pair-breaking $S = -1$, corresponding to full retroreflection, in which case the phase shift $\varphi$ experienced by the retro-reflected quasiparticle is $\varphi = \pi$ for all incoming/outgoing trajectories and surface-bound states accumulate at zero energy, since the excitation energy is given by $E/\Delta = \pm \cos(\varphi/2)$ [28], where $\Delta$ is the gap along the quasiparticle's trajectory (see density of states calculation as a function of specularity in Fig. 3c). However, momentum scattering with a preponderance of retroreflection is inconsistent with measurements of boundary slip in viscous transport in the normal state[32], and incompatible with the atomically smooth silicon surface.

We invoke magnetic surface scattering to explain this stronger $T_c$ suppression in the presence of the magnetic solid $^3$He surface boundary layer. Exchange interaction between quasiparticles and isolated magnetic impurities has been theoretically established to induce additional bound states in superconductors through the Yu–Shiba–Rusinov mechanism[44]. Magnetic scattering by localized $^3$He[45] strongly influences the observed superfluid phase diagram of $^3$He in different aerogels[46,47]. Here we extend these ideas to consider exchange scattering by the uniform 2D surface layer. We seek processes which generate an excess of zero-energy states over that found for diffuse momentum scattering (Fig. 3b and Supplementary Note 6).

The structure of the order parameter is such that the phase shift $\varphi$ experienced by the scattered quasiparticle, and hence the energy of the surface-bound states, will be influenced by spin-dependent scattering processes. To account for extra pair breaking we need to include quantum spin dynamics of randomly oriented localized quantum spins, allowing for their spin flips. Exchange coupling

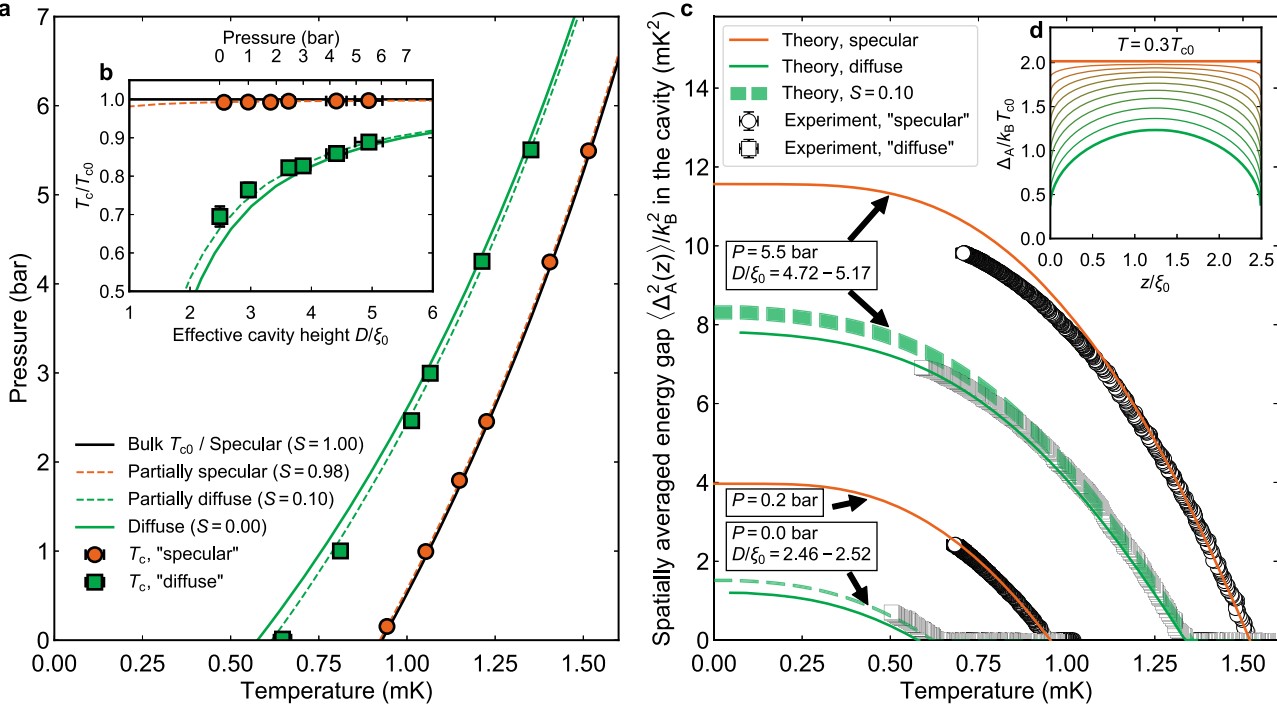

**Fig. 2 Suppression of superfluid transition temperature and superfluid gap for different surface scattering conditions. a** Measured pressure dependence of $T_c$ for close to diffuse (squares) and close to specular (circles) boundary conditions. Full lines show predicted $T_c$ for diffuse (green) and fully specular (black) boundaries, dashed lines are best fits yielding $S = 0.1$ and $S = 0.98$. **b** Suppression of $T_c$ relative to bulk superfluid transition temperature $T_{c0}$ for 'diffuse' boundary steeply increases with confinement. Suppression of $T_c$ for 'specular' boundary is essentially eliminated. The uncertainties in temperature determination and the modelled distortion of cavity height by pressure are used to define the vertical and horizontal error bars, respectively ('Methods'). **c** Spatial average of energy gap $\Delta_A(z)$, where $z$ is the vertical position in the cavity, inferred from measured frequency shift (Supplementary Note 1), for 'diffuse' (squares) and 'specular' (circles) boundary conditions. All theoretical curves include strong-coupling corrections valid near $T_c$. The 'diffuse' experiments agree best with theory for $S = 0.1$ (dashed lines, see also Supplementary Fig. 9). The emergent discrepancy between theory and experiment at lower temperatures at 5.5 bar, for both scattering conditions, is in agreement with the expected temperature dependence of strong-coupling corrections to the gap (Supplementary Note 4). The width of the theoretical curves for $S = 0.1$ accounts for errors associated with the weak pressure dependence and uncertainty of cavity height ('Methods'). The theoretical curves for diffuse boundary condition (solid green lines) correspond to the mean value of $D/\xi_0$ at given pressure, and the theoretical curves for specular boundary condition (solid orange lines) correspond to the bulk energy gaps of $^3$He-A. **d** The calculated gap profile at zero pressure for specularities between 0 and 1 in intervals of 0.1.

between such spins and incident quasiparticles gives rise to interference between the singlet and triplet scattering channels. In our experimental configuration we find this to be the only viable mechanism resulting in enhanced $T_c$ suppression. We find that for a surface with momentum scattering specularity $S$, the suppression of $T_c$ corresponds to an effective specularity between bounds $-S \leq S_{eff} \leq S$, depending on strength of exchange coupling. Thus this process is only detectable for non-diffuse surfaces, but can give rise to $T_c$ suppression exceeding that for a diffuse surface, as observed. To explain the detected $S_{eff} = -0.4$, we propose that the underlying specularity for momentum scattering from the atomically smooth silicon surface with solid $^3$He surface boundary layer should be $S \geq 0.4$.

## Discussion

We first focus on a more detailed discussion of the in situ tuning of surface specularity in the context of prior work. We emphasize that the surfaces in our experiment are close to atomically flat ('Methods'), unlike the surfaces used in earlier work. We propose that this extreme smoothness accounts for the partially specular momentum scattering we infer in the absence of $^4$He preplating. We suggest that the self-assembled $^3$He solid coating at silicon surface does not significantly degrade the smoothness of the liquid–solid boundary.

This contrasts with previous work where diffuse surface scattering ($S = 0$) has been observed in pure $^3$He. Diffuse scattering

was inferred from normal state transverse acoustic impedance experiments where the surface roughness of the quartz crystal was of the order of the wavelength of visible light[40]. Suppression of the superfluid $T_c$ of saturated films of pure $^3$He was observed using a torsional oscillator, with typical surface roughness of the order of 50 nm[48]. The results agreed with the prediction for fully diffuse surface scattering. This is consistent with our model in which spin-flip magnetic scattering is ineffective when $S = 0$. On the other hand, torsional oscillator studies[33] of surface slip in the normal state with a polished silicon surface of roughness 2 nm found a higher specularity $S \approx 0.2$, confirming the importance of surface roughness.

Our determination of a specularity $S = 0.1$, from superfluid gap and $T_c$ suppression, at the chosen $^4$He coverage of 32 μmol m$^{-2}$ ('Methods') is in qualitative agreement with prior normal state measurements[33,35]. These established a rapid increase in specularity for $^4$He coverages greater than around 2 layers, attributable to the required coverage for the onset of superfluidity in a $^4$He surface film with $^3$He overlayer[49]. The absolute values of specularity inferred in the normal state are subject to uncertainty due to imperfect agreement with normal state slip theory[33]. We also note the difference between treatment of surface scattering in kinetic transport theory[50], via phenomenological scattering rates for distribution functions, and quasiclassical theory of the superfluid[31,41], via a more microscopic surface scattering matrix. This means that a comparison of boundary scattering parameters is not entirely straightforward.

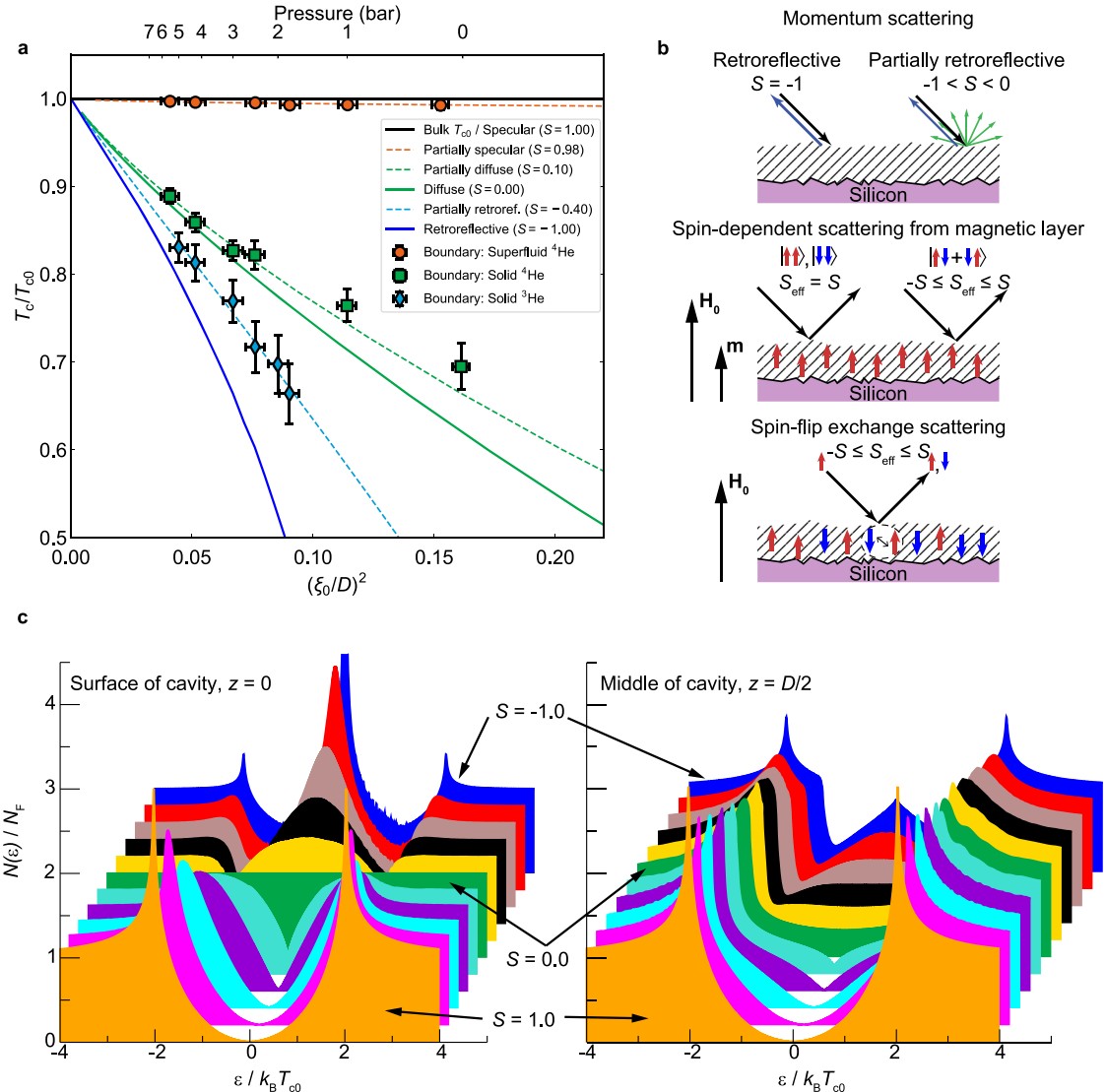

**Fig. 3 Increased suppression of superfluid transition temperature in presence of a magnetic solid $^3$He surface boundary layer. a** Suppression of $T_c$ in cavity relative to that of bulk liquid as a function of square of the inverse effective cavity height $D/\xi_0$, for solid $^3$He boundary (diamonds), solid $^4$He boundary (squares), and superfluid $^4$He boundary (circles). Full lines show: maximal pair-breaking retroreflection ($S = -1$); diffuse ($S = 0$); fully specular ($S = 1$). Dashed lines show best fits to the data: $S = -0.4, 0.1, 0.98$. For the solid $^3$He boundary, $T_c$ is identified from onset of superfluid frequency shift after correcting for background frequency shift arising from magnetic solid layer (Supplementary Note 5). The vertical and horizontal error bars reflect the uncertainties in temperature determination and the height range of the pressure-distorted cavity, respectively ('Methods'). **b** Three candidate scattering mechanisms for negative effective specularity (see also Supplementary Note 6): retroreflection (ruled out by normal state measurements); spin-dependent pair breaking on scattering from a magnetically polarized layer (absent for the relative orientation of surface layer spin polarization **m**, surface normal, and spin orientation of A-phase pairs in our set-up); spin-flip exchange scattering (spin polarization of surface layer can be zero). Here the effective specularity $S_{\rm eff}$ is a parameter characterising combined magnetic and momentum scattering, bounded by the specularity $S$ that would arise from momentum scattering alone. **c** Calculated density of states (DOS) $N(\varepsilon)$, where $\varepsilon$ is the quasiparticle energy, of $^3$He-A at the surface and in the middle of the cavity, scaled with the normal state DOS, $N_{\rm F}$, and averaged over the full Fermi surface. These calculations correspond to pure momentum scattering between specularities $S = -1.0$ and $S = 1.0$, with step size 0.2 between successive colours. The calculations were performed for cavity height $D/\xi_0 = 5$ and temperature $T = 0.2T_{c0}$. The density of zero-energy bound states increases throughout the cavity as the specularity decreases.

The fact that momentum scattering with the pre-plated solid $^4$He surface boundary layer is close to diffuse contrasts with the significant specularity we infer in the pure $^3$He case from our model of magnetic surface scattering. The replacement of surface $^3$He by $^4$He relies on the differential surface binding energy of the two isotopes due to differences in zero-point energy. It appears that a $^4$He coverage of 32 μmol m$^{-2}$, sufficient to displace the $^3$He magnetic surface boundary layer but below that required for surface $^4$He superfluidity ('Methods'), results in a greater surface roughness than the self-assembled solid $^3$He boundary layer. We

suggest that this is attributable to heterogeneity of the surface binding potential across the geometrically smooth silicon surface, which in the presence of the liquid $^3$He overburden gives rise to a non-uniform coating of $^4$He. At sufficiently high $^4$He coverages we establish a superfluid $^4$He film covering the entire surface, which creates an ideally smooth equipotential surface and naturally leads to fully specular scattering conditions.

It is clear that for a given set-up the surface scattering can be tuned in situ in a controlled way. Nevertheless, these results provide motivation for further systematic study of the specularity,

over a fine grid of $^4$He coverages of the surface boundary layer, to investigate the interplay of momentum and magnetic scattering. This would include detailed measurements in the vicinity of the coverage required to fully displace the $^3$He surface boundary layer, and the somewhat higher threshold coverage for the onset of superfluidity in the $^4$He surface boundary layer[36]. Ideally, measurements of superfluid $T_c$ and gap suppression would be coupled with measurements of surface slip in the normal state, and with the development of an independent measurement of surface specularity in the superfluid.

The elimination of gap suppression by specular surfaces we have demonstrated opens the investigation of cavities of arbitrarily small height towards $D << \xi_0$ entering the quasi-2D limit, in which thermal and spin analogues of the Quantum Hall effect are predicted[51,52]. On the other hand, the superfluidity will be completely suppressed in cavities thinner than 100 nm at zero pressure for diffuse scattering, Fig. 2b. Magnetic $^3$He boundaries may stabilize new order parameters under confinement and influence surface spin currents. Precise determination of the gap in this case requires measurements in lower magnetic fields, in order to suppress the solid dipolar shift and increase the superfluid frequency shift. This should be possible using broadband SQUID NMR[53,54].

Future topological superfluid $^3$He mesoscopic devices should provide a new platform for the study of MZMs at well-defined interfaces. The sculpture of the superfluid by confinement will allow the fabrication of hybrid devices based on different $^3$He 'materials', with clean transparent interfaces, Fig. 4. Design of these platforms is supported by quasiclassical theory, which self-consistently describes the spectrum of surface states, gap suppression, and $T_c$ suppression.

In conclusion, our results show experimentally the sensitivity of the superfluid gap suppression and hence ABS to the details of quasiparticle scattering from the surface. A superfluid $^4$He surface boundary layer results in $S = 0.98$, close to fully specular scattering, which eliminates surface ABS in chiral superfluid $^3$He-A. Whether the specularity can be increased further by a thicker $^4$He film, or whether this is a limit determined by solubility of $^3$He in the $^4$He film, should be resolved by future work. A surface boundary layer of solid $^4$He leads to close to diffuse scattering, with a finite density of low energy ABS. These results complement measurements of the transverse acoustic impedance of superfluid $^3$He-B, which are consistent with the density of surface-bound excitations calculated from quasiclassical theory and their dependence on specularity[40].

A magnetic solid $^3$He surface boundary layer gives rise to a greater suppression of $T_c$ than for diffuse scattering and thus further increases the density of low energy ABS. We show that magnetic scattering from this layer can account for the extra pair breaking. This effect requires a degree of specular momentum scattering and disappears when momentum scattering is diffuse.

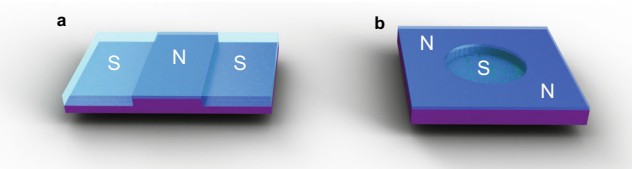

**Fig. 4 Topological mesoscopic superfluidity, where confinement tunes $^3$He into different material phases, enabling hybrid structures. a** An SNS junction, where spatial modulation of cavity height defines SN interfaces. **b** Circular region of higher cavity height defines an isolated mesa of superfluid, cooled through normal liquid in a more confined region. Purple plates represent the silicon wafer, darker blue regions normal liquid $^3$He, and lighter more transparent regions superfluid $^3$He.

We propose that the extremely smooth walls of our nanofluidic geometry are instrumental in this observation. The surface states of $^3$He-A are not topologically protected, unlike its edge states[55] or the Majorana surface states of $^3$He-B[56]. The calculation of the influence of spin-flip magnetic boundary scattering on the density of states of surface excitations in these two topological superfluids remains an open problem.

In more complex topological superconductor architectures designed to realize, detect, and manipulate MZMs for topological quantum computation, it is necessary to eliminate the excess low energy ABS arising from the interface scattering processes such as we have identified here. Meanwhile, our result is a crucial step in the quest to identify and harness Majoranas in liquid $^3$He, the as yet only firmly established topological superconductor/superfluid. More generally, the influence of magnetic degrees of freedom in topological materials[57] and spin-active interfaces in hybrid superconducting-ferromagnetic spintronics[58] are both areas of current interest to which the understanding of spin-dependent surface scattering in a spin-triplet superfluid will contribute.

## Methods

**Silicon nanofluidic cavity fabrication.** The experimental cell was fabricated by direct wafer bonding of two silicon wafers. The confinement region and supporting pillars are defined lithographically on one of the wafers using a process similar to that used in a previous generation of cells[59]. The typical rms surface roughness of the silicon surface is 0.1 nm[60]. This is significantly smoother than the mechanically polished silicon surfaces for which surface specularity has been characterized by normal state studies of slip in viscous transport[32], potentially promoting specularity of surface scattering even in the absence of a superfluid $^4$He film. Deep reactive ion etch (DRIE) is used to create two 300-micron-diameter holes on either side of the confinement region. One acts as a fill line and the other provides a region of bulk helium on the far side of the slab-shaped cavity, Fig. 1a. DRIE is also used to pattern the backside of the wafer to improve the joint between the cell and an external fill line[60]. After all the features are patterned onto the wafers, they are cleaned using a combination of a two-step RCA clean at 75°C followed by immersion in concentrated (49%) HF to remove any oxide or contaminants. The clean patterned wafer is brought into contact with a blank silicon wafer within a wafer aligner, forming a bond between the cavity wafer and the lid. The bond strength is increased and made permanent by an annealing step at 1000°C for 2 h. Successful bonding is confirmed using infra-red imaging and scanning acoustic microscopy. The bonded wafer is diced into individual cells using a diamond saw. A 500-nm-thick silver film is evaporated onto the outside of the bonded wafers to thermalize the cell to the nuclear stage. In order to minimise the effects of differential thermal contraction between the metallic fill line/far-end bulk marker plug and the silicon cell, laser-machined silicon washers are attached around both of the DRIE holes with epoxy (Stycast 1266 mixed with silicon powder), Supplementary Fig. 1. The height of the cavity used in this work was determined to be 192 nm by a profilometer scan on the unbonded wafer. The error in the cavity thickness ±2 nm was estimated from the distribution in height measured in this way across all the cavities on the unbonded wafer. The maximal distortion of cavity height by pressure is determined by finite element method simulations to be 2.6 nm bar$^{-1}$, Supplementary Fig. 2. The horizontal error bars in Figs. 2b and 3a reflect the range in cavity height.

**NMR measurements.** The cooling of the $^3$He within the cell, the thermometry, and the SQUID NMR spectrometer were as used in previous work[24,25], Supplementary Fig. 1. The helium is cooled via the column of $^3$He in the fill line which connects the cell to a sintered silver heat exchanger mounted on a silver plate, connected via a silver rod to the copper nuclear demagnetization stage. A platinum NMR thermometer is mounted on the silver plate. Measurements were made at a $^3$He Larmor frequency of 967 kHz, with the static field of around 30 mT applied along the cavity surface normal ($\hat{z}$). Field gradients were applied to both separate the bulk marker signals from the cavity signal (along $\hat{z}$) and to resolve the signals from the two bulk markers (along $\hat{x}$, $\hat{y}$). The free induction decay following small angle (3–10°) tipping pulses, applied at 10 s intervals, was averaged typically 30 times, and Fourier transformed. To measure the superfluid transitions in the cavity and the bulk markers, the silver plate temperature was swept across the relevant region at a rate below 50 μK h$^{-1}$, Supplementary Fig. 6. The reported values of $T_c$ are averages of several such sweeps. Measurements with tipping pulses of different amplitude enabled a correction to be made for temperature gradients between the $^3$He in the cell and the platinum thermometer (Supplementary Note 7). This correction depended on the surface boundary condition, which influenced the boundary resistance of the silver heat exchanger. The uncertainties in the correction introduce an additional temperature error, which has been included in the error bars of all reported temperature values. The temperature gradient across the

cavity is small, and dependent on surface boundary condition; it is determined from the difference between the measured superfluid transition temperatures in the two bulk marker volumes. For the solid $^4$He and $^3$He surface boundary layers, the difference is around 20 μK, while for the superfluid $^4$He surface boundary layer it is at most 2 μK, Supplementary Fig. 6. This gradient is taken into account in determining the error in superfluid transition temperature.

**In situ tuning of surface scattering**. (i) Magnetic scattering. Pure $^3$He, with $^4$He impurity concentration <100 ppm, is used to fill the empty cell and silver heat exchanger (surface area 8.03 ± 0.04 m$^2$ determined by N$_2$ BET isotherm). This results in a magnetic surface boundary layer of solid $^3$He[43]. As discussed, the results suggest that this self-assembled solid $^3$He layer on the extremely smooth silicon surface gives rise to partially specular momentum scattering which combines with spin-flip magnetic scattering. (ii) Diffuse non-magnetic scattering. In order to displace the naturally occurring magnetic surface boundary layer of $^3$He[43], 32 μmol m$^{-2}$ of $^4$He was added to the empty cell and silver heat exchanger at 30 K, followed by cooling to below 1 K over 30 h, and a subsequent anneal at 2 K for several hours. This coverage is below that established to be necessary to see a superfluid transition in the surface $^4$He layer on a mylar substrate, in the presence of an overburden of $^3$He at saturated vapour pressure[49], and was motivated by previous work[36]. The sample is cooled to 100 mK before adding $^3$He. Under these conditions, the $^3$He surface magnetism seen in pure $^3$He samples is eliminated, as confirmed by the absence of an observable temperature-dependent susceptibility down to the lowest temperatures achieved. We find that the remaining momentum scattering is close to diffuse. (iii) Specular scattering. To create specular scattering conditions from the previous $^4$He surface plating conditions, the cell is pumped at 1.5 K, leaving a residual solid $^4$He 'layer' on the surfaces. Then more $^4$He is added into the cell/heat exchanger. Subsequently, the helium pumped out in the previous step is restored. The sample is slowly cooled into the mK range, during which all the $^4$He forms a surface film of solid $^4$He with a superfluid $^4$He overlayer. With nominal surface $^4$He coverage in the range 68–139 μmol m$^{-2}$, we always detect the same specularity, consistent with previous work[32], which found evidence for surface scattering close to specular for surface film coverages in excess of 60 μmol m$^{-2}$. The quoted surface coverages assume that the thin $^4$He films coat the heat exchanger and sample volume uniformly. We note that differences in heat exchanger structures and surface area determination, as well as varying methodologies for adding $^4$He into the system may lead to systematic differences between nominal sample surface $^4$He coverages between different set-ups in different laboratories.

**Theoretical calculation of gap suppression**. Theoretical calculation of gap suppression is made using quasiclassical theory that can systematically include spatial variations of the order parameter and Andreev scattering process. The boundary conditions on the propagator are incorporated using the random S-matrix scattering model (Supplementary Note 3).

## Data availability

The $T_c$ and gap suppression data and calculations that support the findings of this study are available in Figshare, https://doi.org/10.17637/rh.12777620[61].

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

## Acknowledgements

We thank M. Eschrig, J.A. Sauls, and S. Simon for helpful discussions. This work was supported by EPSRC grants EP/J022004/1 and EP/R04533X/1, NSF grants DMR-1708341 and DMR-2002692, and the European Union's Horizon 2020 Research and Innovation Programme, under Grant Agreement no. 824109 (European Microkelvin Platform). Fabrication was carried out at the Cornell Nanofabrication Facility (CNF) with assistance and advice from technical staff. CNF is supported in part by the NSF through ECCS-1542081. Measurements we made at the London Low Temperature Laboratory, supported by technical staff, in particular Richard Elsom, Ian Higgs, Paul Bamford, and Harpal Sandhu.

## Author contributions

Experimental work was carried out by P.J.H. and L.V.L. with contributions from A.C. The nanofabricated cells were prepared and assembled by N.Z., X.R. and A.C. X.R. carried out the FEM simulations of the cell. The analysis and presentation was carried out by P.J.H., L.V.L. and A.V. with contributions from J.S. A.V. performed calculations of gap profile and superfluid transition temperature. A.V developed the theory of magnetic scattering with contributions from P.S. The work at Cornell was supervised by J.M.P., and the work in London was supervised by J.S., A.C. and L.V.L., who had the leading roles in formulating the research. P.J.H., J.S. and A.V. had leading roles in writing the paper, with contributions from all authors.

## Competing interests

The authors declare no competing interests.
