## [Peer Review File · Nature Communications]

REVIEWER COMMENTS

Reviewer #1 (Remarks to the Author):

The manuscript "Fragility of surface states in topological superfluid ^3He " by P.J. Heikkinen et al. reports on superfluid ^3He in a record-breaking narrow gap with very smooth surfaces at different ^4He coverages. The superfluid properties are investigated in detail, using pure ^3He and ^4He pre-plated surfaces. The authors are experts in this field, where they have made important contributions.

The main conclusion of the manuscript ("We discover that surface magnetic scattering leads to unexpectedly large suppression of T_c") is based on the comparison of experimental results with theoretical calculations. For the first time, a model for magnetic scattering for ^3He in a narrow gap is developed, although it contains fitting parameters. The results are new and important, and I think that the manuscript is worth being published in Nature Communications. However, I believe that the main conclusion, should be formulated as an assumption, not an assertion. This opinion is based on the following questions which I ask to clarify:

1. In previous experiments it was found that in pure ^3He the scattering is diffuse and there is an abrupt transition to nearly specular scattering when ^4He is added. For example, in Ref.(32) it was found that at the coverage of $20.8 \mu\text{mol}/\text{m}^2$ $S = 0$, at the coverage of 30 $S=0.6$ and $S=0.9$ already at the coverage of 39.2. The manuscript states that $S=0.1$ at the coverage of 32, although the surfaces are much smoother than in Ref.(32). How is this possible? I expect that $32 \mu\text{mol}/\text{m}^2$ should result in $S>0.6$ even for a rougher surface. Perhaps the point is the inaccuracy of the theoretical model, but then it cannot be used to determine S from the experimental data. It may be the task for the future to measure S with the same coverage independently, for example, by measuring the spin diffusion in the normal phase.
2. In all previous experiments in pure ^3He , it was shown that S is nearly 0. The small amount of ^4He did not change S , and starting from about $25 \mu\text{mol}/\text{m}^2$ the S value rapidly increases. The main conclusion about the effect of magnetic scattering is based on the theoretical model described in the Supplementary Note 6. It was noted that the model is applicable only if in pure ^3He $S>0.4$. The authors believe that this is possible, since their surface is almost perfectly smooth. But this is only assumption and it remains unclear why $32 \mu\text{mol}/\text{m}^2$ of ^4He reduces S to 0.1.

Reviewer #2 (Remarks to the Author):

The authors have developed a unique experimental setup that allows them to study the superfluid phases of ^3He in a small cavity with a thickness of only a few coherence lengths. In this article, they have used this system to carry out careful experiments on the effect of surface scattering on the superfluid A-phase, using NMR to measure the suppression of the transition temperature and energy gap. Due to their well-defined geometry, the authors can analyze these results using quasiclassical theory, which describes the experiments well, and allows the authors to draw conclusions about the surface scattering and calculate the density of surface states. The authors show that the large T_c suppression in the presence of magnetic solid ^3He can be explained by magnetic scattering processes, which further increase the density of low energy surface states. This represents the next step in understanding the effects of magnetic scattering on superfluid ^3He , where, as the authors mention, several recent papers have found surprising results in the presence of scattering from magnetic

impurities. Due to their well-defined geometry, the authors can draw more detailed conclusions about this magnetic scattering process.

In addition to the significance within the field of superfluid ^3He , these results have relevance to the broader field of condensed matter physics. As the authors point out, ^3He is the only established topological superfluid with many ongoing searches for topological superconductors for use in quantum computation, in which the surface states are particularly relevant. ^3He provides a unique system in which boundary conditions can be tuned in situ to tune and study the surface states, as the authors have demonstrated. The results here can contribute to better understanding of other topological superconductors, particularly how interface details, such as magnetization, can affect the order parameter and surface bound states.

The paper is well written, and I recommend it for publication in Nature Communications.

I have a few optional suggestions which I believe would improve clarity for non-experts:

1) The authors show the calculated density of states at the surface in Supplementary Fig. 8. Since an emphasis of the article is the increase in zero energy states as surface scattering changes from specular to diffuse to magnetic, I would suggest showing the surface density of states in the main text for the relevant values of specularity so that they can be more easily be referred to.

2) I suggest that the authors point out that diffuse scattering leads to a finite density of low energy Andreev bound states earlier in the article, perhaps in paragraph 6 where it is noted that these states are not present with specular scattering.

(Changes made to the manuscript are highlighted in bold.)

The authors would like to thank both reviewers for their positive comments and helpful suggestions. We have reformatted the manuscript, which was originally submitted to Nature Physics, by introducing Introduction, Results and Discussion sections appropriate for Nature Communications. We have addressed all points, highlighting the major changes in text in blue in the revised manuscript.

The material on earlier studies of superfluid ^3He in the Introduction is expanded somewhat.

Change to manuscript: add penultimate paragraph in Introduction.

In the Discussion section we add more text on the tuning of surface specularly, including a comparison with previous work. This clarifies issues raised by Referee 1 (see also the response below), of particular interest to ^3He specialists. We have tried to structure the Discussion section so that key points that should be of interest to a wide readership are not obscured. Overall we believe that the manuscript is strengthened.

Change to manuscript: add an expanded discussion in the Discussion section.

We have also taken the opportunity to make some minor modifications to the Figures to improve clarity of presentation. We have also added Supplementary Figure 9 for completeness to show the energy gap suppression at various pressures with “diffuse” boundary condition.

Reviewer 1

Reviewer 1 has two concerns. The first is about the precise relationship between specularly and coverage of the surface ^4He boundary layer. The second concerns our proposal that in our experiment with pure ^3He we require significant specularly for momentum scattering, in contrast to the Referee's belief that “in all previous experiments in pure ^3He , it was shown that S is nearly 0”. In summary we have revised the manuscript to clarify these matters and address these concerns, see discussion below. We note that our experiment was performed with a few carefully chosen ^4He surface platings, and we better explain the rationale behind this choice, and its context, in the revised manuscript.

Changes to manuscript:

First five paragraphs of Discussion section + added discussion in “In situ tuning of surface scattering” section of Methodology. Modification of Fig. 1d.

We respond to the Referee's points as follows:

1. In previous experiments it was found that in pure ^3He the scattering is diffuse and there is an abrupt transition to nearly specular scattering when ^4He is added. For example, in Ref.(32) it was found that at the coverage of $20.8 \mu\text{mol}/\text{m}^2$ $S = 0$, at the coverage of $30 S=0.6$ and $S=0.9$ already at the coverage of 39.2 . The manuscript states that $S=0.1$ at the coverage of 32 , although the surfaces are much smoother than in Ref.(32). How is this possible? I expect that $32 \mu\text{mol}/\text{m}^2$ should result in $S>0.6$ even for a rougher surface. Perhaps the point is the inaccuracy of the theoretical model, but then it cannot be used to determine S from the experimental data. It may be the task for the future to measure S with the same coverage independently, for example, by measuring the spin diffusion in the normal phase.

Indeed, previous work has demonstrated a rapid increase in specularly, with onset at a particular ^4He surface coverage. The precise identification of this coverage was beyond the scope of our experiment, and is the quantity most subject to systematic error. It is important to note that the results of old Ref. [32] are discussed, by the same authors (Tholen and Parpia), in more detail in Phys. Rev. B **47**, 319

(1993). These are now Refs. [32,33] in the current manuscript. The longer paper makes clear that the quantitative results of this work are subject to uncertainty. There are two sources of uncertainty: the process to infer specularly from measurements of slip in the normal state; uncertainty in the nominal values of surface helium coverage.

Our experiment was performed with a few carefully chosen ^4He surface platings. Our choice of $32 \mu\text{mol}/\text{m}^2$ is now referred to explicitly on p5, with a reference to **Methods**, where more detail is supplied. We can empirically establish that this ^4He plating eliminates the solid ^3He boundary layer. We rely on other work, referred to in **Methods**, which shows that this coverage (on Mylar) is below that required for the onset of superfluidity in the surface ^4He film, with an overburden of ^3He . At this coverage we demonstrate that both T_c and gap suppression are accounted for by $S=0.1$, within the framework of quasiclassical theory.

In more detail:

- (i) Inferences of the relationship between specularly and surface ^4He coverage from different experiments are subject to uncertainties in latter which are difficult to quantify.

This point is discussed in Methods. Comparisons of reported values of ^4He coverage between different experimental set-ups need to be made carefully. In all cases ^3He sample is cooled using sintered metal powder heat exchanger, the surface area of which dominates over the sample area. Thus, to determine the surface coverage in the sample container after preplating with ^4He , we need to establish surface area of the heat exchanger. Further, we assume that the thin helium films of few atomic layers coat the heat exchanger and the sample volume to the same thickness. Differences in heat exchanger structures, surface area determination, and methodologies for adding ^4He into the system could lead to systematic variations between sample surface ^4He coverages between different set-ups in different laboratories. The main observable which this affects is the nominal ^4He coverage at which there is a sudden onset of an increase in specularly, which is the focus of the Referee's comment.

- (ii) Inferred values of S arise from different underlying theories, for the normal and superfluid states respectively.

This point is mentioned in third paragraph of Discussion.

In Ref. [32,33] of present manuscript, the specularly is inferred from the normal state slip length based on kinetic theory describing the transport and scattering of quasiparticles (see for example J. Low Temp. Phys. **53**, 695 (1983)). This is imperfect since it is found that the specularly must be allowed to be temperature dependent, contrary to expectations, to conform to slip theory, Fig. 5 of Ref. [33]. In our work, we use more microscopic quasiclassical theory of superfluid ^3He to model the scattering boundary conditions (Supplementary Note 3). This theory is more appropriate in describing the pair-breaking effects that are the most prominent feature in our system. There is some discussion of two formulations for calculating the transverse acoustic impedance (quasiparticle kinetic equation vs quasiclassical theory) in J. Phys. Soc. Jpn. **77**, 111003 (2008), **which we now include as Ref. [41] in our manuscript.**

2. In all previous experiments in pure ^3He , it was shown that S is nearly 0. The small amount of ^4He did not change S , and starting from about $25 \mu\text{mol}/\text{m}^2$ the S value rapidly increases. The main conclusion about the effect of magnetic scattering is based on the theoretical model described in the Supplementary Note 6. It was noted that the model is applicable only if in pure ^3He $S > 0.4$. The authors believe that this is possible, since their surface is almost perfectly smooth. But this is only assumption and it remains unclear why $32 \mu\text{mol}/\text{m}^2$ of ^4He reduces S to 0.1.

We agree with the Referee that in many earlier experiments with pure superfluid ^3He the specularly has been measured to be close to diffuse. **We include an explicit discussion of this point on pages 8 and 9.** As the Referee recognises, we believe the surface roughness plays a key role. Prior experiments commonly have significantly rougher surfaces (such as polished metal or glass) compared to atomically smooth silicon surface in our work. We include a discussion of these results. However, there is one important exception. This is the previous work on a relatively smooth silicon surface (but with a quoted surface roughness of 2 nm on a horizontal scale of 50 μm , so rougher than our surface) by Tholen and Parpia, Ref. [32,33]. This work reports a specularly of approximately $S = 0.2$ [normal state analysis, subject to caveats discussed above]. Assuming the non-zero specularly results from the smooth silicon surface supports our hypothesis of yet higher specularly for momentum scattering from our smoother surface. This was previously referred to in the Methods section; **we now include an expanded version in the Discussion, where we explicitly refer to the surface roughness in previous work.**

We suggest there that the solid ^3He layer, on the extremely smooth silicon surface in our cell, self-assembles and preserves the underlying smoothness. This would allow partially specular momentum scattering, even $S > 0.4$. When combined with spin-flip magnetic scattering, as described in Supplementary Note 6, this can result in effective specularly $-S < S_{\text{eff}} < S$ and more than diffuse T_c suppression.

The reviewer questions why $32 \mu\text{mol}/\text{m}^2$ of ^4He reduces S to 0.1. To address this point in the previously submitted manuscript we wrote (Methods section). “The results find that the remaining momentum scattering is close to diffuse, suggesting that the ^4He pre-plating surface layer has a greater roughness than the self-assembled solid ^3He boundary layer occurring in pure ^3He sample”. **We now include expanded text in paragraph 4 of the Discussion.**

We have redrafted the penultimate conclusion paragraph to address the Referee’s point “the main conclusion should be formulated as an assumption, not an assertion”. Here it is stated as a reasonable theoretically-motivated hypothesis to explain our observations, given the exceptional smoothness of our silicon surface.

Reviewer 2

1. The authors show the calculated density of states at the surface in Supplementary Fig. 8. Since an emphasis of the article is the increase in zero energy states as surface scattering changes from specular to diffuse to magnetic, I would suggest showing the surface density of states in the main text for the relevant values of specularly so that they can be more easily be referred to.

We have incorporated this change and added the plot of calculated density of states as Fig. 3c. The Supplementary Note 3 has been modified accordingly.

2. I suggest that the authors point out that diffuse scattering leads to a finite density of low energy Andreev bound states earlier in the article, perhaps in paragraph 6 where it is noted that these states are not present with specular scattering.

The text of the final paragraph of the Introduction has been modified accordingly.

REVIEWERS' COMMENTS

Reviewer #1 (Remarks to the Author):

I agree that the ^4He pre-plating surface layer may have a greater roughness than the self-assembled solid ^3He boundary layer occurring in pure ^3He sample on a very smooth surface, although additional independent measurements of the surface specularity are required to confirm this. It is not easy, so this is a challenge for the future. The results described in the manuscript will undoubtedly stimulate such investigations.

I am convinced by the additional explanations given, and fully satisfied by the changes made in this resubmission. Overall, the authors have made the manuscript much more accessible and I strongly recommend publication of this manuscript in Nature Communications.

Reviewer #2 (Remarks to the Author):

The authors have improved their original manuscript and addressed my previous comments. They have added more detailed introduction and discussion sections that put the work into a broader context with detailed comparisons to previous work and I believe address the concerns raised by the other reviewer.

The work represents an important step forward in engineering the surface states of superfluid ^3He and I believe it should be published in Nature Communications.